# Metabolic Regulation of Mitochondrial Protein Biogenesis from a Neuronal Perspective

**DOI:** 10.3390/biom12111595

**Published:** 2022-10-29

**Authors:** Jara Tabitha Hees, Angelika Bettina Harbauer

**Affiliations:** 1TUM Medical Graduate Center, Technical University of Munich, 81675 Munich, Germany; 2Max Planck Institute for Biological Intelligence, in Foundation, 82152 Planegg-Martinsried, Germany; 3Institute of Neuronal Cell Biology, Technical University of Munich, 80802 Munich, Germany; 4Munich Cluster for Systems Neurology, 81377 Munich, Germany

**Keywords:** PGC-1α, mitochondrial biogenesis, transcription, translation, AMPK, mTORC1, insulin, neurons

## Abstract

Neurons critically depend on mitochondria for ATP production and Ca^2+^ buffering. They are highly compartmentalized cells and therefore a finely tuned mitochondrial network constantly adapting to the local requirements is necessary. For neuronal maintenance, old or damaged mitochondria need to be degraded, while the functional mitochondrial pool needs to be replenished with freshly synthesized components. Mitochondrial biogenesis is known to be primarily regulated via the PGC-1α-NRF1/2-TFAM pathway at the transcriptional level. However, while transcriptional regulation of mitochondrial genes can change the global mitochondrial content in neurons, it does not explain how a morphologically complex cell such as a neuron adapts to local differences in mitochondrial demand. In this review, we discuss regulatory mechanisms controlling mitochondrial biogenesis thereby making a case for differential regulation at the transcriptional and translational level. In neurons, additional regulation can occur due to the axonal localization of mRNAs encoding mitochondrial proteins. Hitchhiking of mRNAs on organelles including mitochondria as well as contact site formation between mitochondria and endolysosomes are required for local mitochondrial biogenesis in axons linking defects in any of these organelles to the mitochondrial dysfunction seen in various neurological disorders.

## 1. Introduction

Mitochondria are double membrane organelles that harbor their own DNA (mtDNA). They are often referred to as the powerhouse of the cell as they produce adenosine triphosphate (ATP) via oxidative phosphorylation (OXPHOS). The OXPHOS pathway generates ATP by several oxidation-reduction reactions involving electron transfer from NADH and FADH_2_ to oxygen across transmembrane protein complexes in the inner mitochondrial membrane. In these reactions, NADH and FADH_2_ are oxidized to NAD^+^ and FAD, respectively [1]. Apart from ATP production, mitochondria also play essential roles in various other cellular functions such as intracellular calcium (Ca^2+^) homeostasis and regulation of apoptosis [2]. Neurons critically depend on mitochondria due to their high energy demand and need for tight Ca^2+^ regulation to maintain neuronal activity [3]. Consequently, mitochondrial dysfunction is directly linked to neurodegenerative diseases and aging [4,5]. Neurons are highly polarized cells with a complex structure and different subcellular compartments including a cell body, dendrites, axons and synapses. The different parts of the neuron do not only have different functions but also different energy demands. As a consequence, a finely tuned mitochondrial network constantly adapting to the local requirement of the neuronal compartment is necessary. This is achieved by mitochondrial transport along axons and dendrites, fusion and fission and degradation of damaged organelles via mitophagy, as well as mitochondrial biogenesis [6,7]. As mitochondria cannot be made de novo, the biogenesis of mitochondrial proteins requires already existing organelles, which duplicate and express their mtDNA, as well as import of over 1000 proteins encoded in the nucleus [8]. Hence, a tight regulation and constant crosstalk between the nucleus and mitochondria is required to ensure proper mitochondrial biogenesis.

## 2. Regulation of Mitochondrial Biogenesis by PGC-1α

The co-transcriptional regulation factor PGC-1α (peroxisome-proliferator-activated γ co-activator-1α) is known as the master regulator of mitochondrial biogenesis. It activates different transcription factors including NRF-1 and NRF-2 (nuclear respiration factors 1 and 2), which promote the expression of several nuclear-encoded mitochondrial genes. Furthermore, they drive the expression of the nuclear-encoded TFAM (mitochondrial transcription factor A), which is required for transcription and replication of mtDNA [9,10].

Several intracellular signaling molecules generated or regulated by mitochondria (Figure 1) are involved in promoting mitochondrial biogenesis via the PGC-1α-NRF-1/2-TFAM pathway, including the AMP/ATP ratio (via AMPK), the NAD^+^/NADH ratio (via SIRT1) and Ca^2+^ levels (via CaMK) [11]. AMPK (adenosine monophosphate-activated protein kinase) is a serine/threonine protein kinase that is composed of one catalytic α subunit and two regulatory β and γ subunits [12]. An increased AMP/ATP ratio induces activation of AMPK, which directly phosphorylates and thereby activates PGC-1α [13]. PGC-1α in turn controls expression of several mitochondrial genes as well as its own [13,14]. In addition to activating AMPK, AMP can also be converted to cyclic AMP (cAMP) by adenyl cyclase and subsequently stimulate PKA activity. PKA in turn phosphorylates the cAMP response element binding (CREB) protein in the nucleus [15]. CREB is a transcription factor that binds to the promoter of PGC-1α [16], thereby promoting mitochondrial biogenesis. Apart from AMPK, SIRT1 (Sirtuin 1) is another energy sensor that plays an important role in mitochondrial biogenesis. SIRT1 is a deacetylase that is activated in response to an increased NAD^+^/NADH ratio caused by energy stress. Once activated, SIRT1 deacetylates PGC-1α resulting in its activation [17]. Interestingly, the signaling pathways of the two major energy sensors, AMPK and SIRT1, seem to be interconnected as AMPK acts upstream of SIRT1 by increasing intracellular NAD^+^ levels leading to SIRT1 activation, deacetylation of PGC-1α and mitochondrial biogenesis [18,19,20,21]. Finally, Ca^2+^ also plays an important role in regulating mitochondrial biogenesis via the PGC-1α-NRF-1/2-TFAM pathway [22]. Mechanistically, Ca^2+^ promotes CaMK (calcium/calmodulin-dependent protein kinase) activity, which phosphorylates p38 MAPK (p38 mitogen-activated protein kinase), resulting in activation of PGC-1α [22,23,24]. Interestingly, CaMK can also stimulate PGC-1α via CREB activation [14]. Consequently, CREB might be involved in both AMP- and Ca^2+^-dependent mitochondrial biogenesis.

Since the majority of the studies about mitochondrial biogenesis have been performed in non-neuronal cells, very little is known about its regulation in neurons. However, as in other cells, PGC-1α is the master regulator of mitochondrial biogenesis in neurons [11]. Overexpression of PGC-1α has been shown to result in an increased number of mitochondria and improved mitochondrial function in primary hippocampal neurons [25]. Knockdown of PGC-1α, on the other hand, leads to a reduction in dendritic mitochondria and inhibition of synaptogenesis in hippocampal neurons [26]. Furthermore, PGC-1α has been shown to control axonal mitochondrial density in a SIRT1-dependent manner [27]. It remains to be determined how much local activation of AMPK, SIRT1 or CaMK can elicit global changes in mitochondrial transcription as discussed below.

## 3. Regulation of Mitochondrial Biogenesis by Signaling Pathways

Insulin, the main hormone involved in fuel metabolism, is an important regulator of mitochondrial biogenesis (Figure 1). Upon binding to the insulin receptor, insulin elicits the phosphoinositide 3-kinase (PI3K)-dependent activation of AKT [28]. AKT in turn phosphorylates and thereby inhibits the transcription factor FOXO1 (Forkhead Box Protein O1) [29,30,31]. AKT-induced FOXO1 inhibition increases expression and activity of PGC-1α thereby stimulating transcription of mitochondrial genes [32,33,34,35,36]. Despite some conflicting studies [37,38], the overall agreement is that insulin signaling promotes mitochondrial biogenesis [39,40]. In line with this, several studies have shown that insulin treatment stimulates mitochondrial protein synthesis and function [41,42]. AKT also activates mTORC1 (mammalian target of rapamycin complex 1), which not only regulates mitochondrial oxidative function [43] via stimulation of PGC-1α [44] but also increases the translation of nuclear-encoded mitochondrial proteins via activation of eIF4E (eukaryotic translation initiation factor 4E) [45]. Interestingly, insulin signaling is also clearly linked to various aspects of neuronal mitochondrial function including respiration, ATP production and Ca^2+^ buffering as well as protein homeostasis and biogenesis [46]. In cortical neurons, insulin treatment has been shown to enhance mitochondrial respiration [47]. This in vitro finding could be confirmed in mice, where intranasal application of insulin increases brain mitochondrial respiration accompanied by enhanced mtDNA levels as well as increased mitochondrial protein levels and PGC-1α expression in the hippocampus [48]. This supports the model that insulin signaling promotes mitochondrial biogenesis also in neurons.

To add to the complexity, the insulin/AKT pathway has also been shown to inhibit AMPK through phosphorylation via AKT [49,50,51,52,53,54,55]; and mTORC1 directly inhibits AMPK through an inhibitory phosphorylation [56]. AMPK, conversely, can also downregulate mTORC1 signaling via phosphorylation [57,58]. This reciprocal inhibition may result in a differential regulation of mitochondrial biogenesis at the transcriptional versus the translational level. Hence inhibition of mTORC1 signaling by AMPK would reduce mitochondrial protein biogenesis despite an increase in transcription and vice versa. This may explain how mitochondrial synthesis and activity can be decreased (via reduced mTORC1 signaling) despite increased expression of PGC-1α (via increased AMPK signaling). This is in line with several studies demonstrating that the correlation between mammalian expression levels of mRNA and protein is fairly low [59,60,61] and that shifts in metabolism, as for example seen during neuronal differentiation, are executed at the translational level [62].

Of note, the signals stimulating PGC-1α expression, including the AMP/ATP ratio, the NAD^+^/NADH ratio and Ca^2+^, are part of mitochondrial feedback mechanisms within the cell (Figure 1). Cell-intrinsic signals may demand a more long-term and therefore slower adaptation, which is obtained by transcriptional upregulation of mitochondrial genes, in contrast to the quick and transient regulation of translation by mTORC1 triggered by extrinsic signals such as insulin and other growth factors.

Finally, also other steps beyond transcription and translation could be controlled by cellular signaling pathways. This includes the localization of mRNAs encoding mitochondrial proteins (as discussed below) and the regulated removal of mitochondria by selective autophagy processes (mitophagy). In mammals, several mitophagy pathways and proteins have been described including PINK1 (PTEN-induced kinase 1), Parkin, BNIP3L/NIX and FUNDC1 [63]. The PINK1/Parkin-dependent pathway is the best characterized pathway for degradation of damaged mitochondria. Briefly, in healthy mitochondria, PINK1 needs to be constantly synthesized, imported and degraded [64]. On damaged mitochondria, however, PINK1 is stabilized and recruits Parkin to the mitochondria, which in turn leads to initiation of mitophagy, recruitment of autophagy receptors and eventually lysosomal degradation [65,66]. Similar to mitochondrial biogenesis, mitophagy is also regulated by cellular signaling pathways including AMPK signaling (see [67] for more detail). Interestingly, in addition to their well-known role in mitophagy, PINK1 and Parkin are also involved in regulating mitochondrial biogenesis (as discussed below). Additionally, the import of mitochondrial precursor proteins through the translocases of the outer (TOM) and inner membrane was found to be regulated by several kinases in yeast [8,68,69,70,71] and recently also confirmed to be regulated in mammalian cells [72]. It will be interesting to reveal further mechanistic connections between these processes and the transcriptional as well as translational pathways described here.

## 4. Mitochondrial Biogenesis by mRNA Localization and Axonal Translation

As the majority of mitochondrial proteins is encoded in the nucleus, it has long been assumed that mitochondrial biogenesis is restricted to the neuronal cell body. Newly generated mitochondria subsequently travel to distal parts of the neurons and replace damaged mitochondria, which in turn are retrogradely transported to the cell body for degradation. The speed of transport, however, which is estimated to be around 0.5 µm/s in neurons [73,74], creates a challenge for the neuron since newly generated mitochondria would take days to reach distal parts of the axon. While mitochondrial proteins are generally longer lived than other mammalian proteins [75,76], there are also exceptions to this rule. In line with this, nuclear-encoded mitochondrial transcripts have been found in axons [77,78,79,80], and, interestingly, transcripts encoding for mitochondrial proteins have been shown to be significantly enriched in axons compared to the somatodendritic region [81,82]. Furthermore, translation of both mitochondrial- [83] and nuclear-encoded [84,85,86] mitochondrial proteins could be observed in axons.

mRNAs contain cis- and trans-acting factors, which determine their subcellular localization. Some mRNAs have 3′ untranslated region (UTR) motifs, called ‘zipcodes’, which regulate the localization of the mRNAs to axons [87,88,89]. Additionally, mRNAs can interact with RNA-binding proteins (RBPs) forming so-called messenger ribonucleoprotein (mRNP) granules. These assemblies are important for regulation of subcellular transport and stability of mRNAs. RNPs can phase-separate into membraneless foci [90] and interact with motor proteins for transport along axons and dendrites [91,92]. One example is SFPQ (splicing factor, proline-glutamine rich), an RBP that binds multiple mRNAs, including *laminb2* and *bclw*, allowing for their trafficking along axons [93] (Figure 2A). Both *laminb2* and *bclw* transcripts have been shown to be locally translated in axons and their protein products localize to mitochondria [86,94]. An emerging concept is the tethering of mRNPs directly onto organelles allowing for mRNA transport along axons and also on-demand local translation in distal parts of the neurons. It has already been demonstrated that mRNPs can hitchhike on mitochondria, early endosomes, late endosomes and lysosomes [84,85,95,96,97]. Since these organelles are being transported back and forth along neurons, hitchhiking represents an ideal, energy efficient way to distribute mRNAs in axons. We have recently shown that the transcript encoding for the mitochondrial protein PINK1 and potentially also other nuclear-encoded mitochondrial transcripts are tethered to neuronal mitochondria allowing for axonal co-transport [85] (Figure 2A). PINK1, one of the main players in mitophagy, is a mitochondrial protein with a very short half-life [98,99]. Interestingly, mitochondrial hitchhiking of the *Pink1* mRNA requires translation of the protein in addition to binding of the transcript. Furthermore, the coding region rather than the 3′UTR is required for mitochondrial localization [85]. The tethering complex for the *Pink1* transcript is composed of the outer mitochondrial membrane protein Synaptojanin 2 binding protein (SYNJ2BP) and Synaptojanin 2 (SYNJ2), which contains an RNA-binding motif [85] (Figure 2A). Interestingly, SYNJ2BP itself has also been identified as an RBP [100,101]. A similar translation dependent mechanism may be tethering the *Cox7c* (cytochrome c oxidase subunit 7C) transcript to mitochondria. *Cox7c* mRNA encoding an essential component of the mitochondrial respiratory chain has been demonstrated to be associated and co-transported with mitochondria along axons [95] (Figure 2A). Another recent study has found several mRNAs localized to early endosomes [102]. Since part of the mRNAs dissociate from the early endosomes upon puromycin treatment, the localization can be either translation-dependent or -independent [102]. Furthermore, for *EEA1* (early endosomal antigen 1) mRNA encoding an endosomal tethering factor and fusogen, the coding sequence is sufficient for endosomal localization, while the 3′UTR is not required [102] (Figure 2B), similar to the *Pink1* and *Cox7c* transcript tethering. Fittingly, early endosomes and endolysosomes as well as mitochondria have been shown to be hotspots of local protein synthesis in axons [84,97,103].

How ribosomes are localized to those specific hotspots is an active area of research. A recent study identified the novel Rab5 effector complex FERRY in neurons, which localizes to early endosomes and interacts both with the translation machinery and mRNAs [97]. Importantly, it has been shown to selectively bind to mRNAs that are enriched for nuclear-encoded mitochondrial genes and to colocalize with mitochondria [97] (Figure 2B), suggesting that the FERRY complex transports ribosomes involved in local mitochondrial biogenesis.

Finally, also hitchhiking of RNA granules on endolysosomes has been reported. RNPs containing nuclear-encoded mitochondrial transcripts associate with moving Rab7-positive endolysosomes in axons [84]. Moreover, annexin A11 (ANXA11) was identified as the tether that links G3BP1 (GTPase-activating protein SH3 domain-binding protein 1)-containing RNA granules to endolysosomes [96]. While the RBP G3BP1 can be part of stress granules, where translationally stalled RNAs are stored upon stress [104], G3BP1 can also localize to RNA granules positive for CLUH (clustered mitochondria homologue) distinct from stress granules [105]. CLUH has been identified as an RBP that binds a subset of nuclear transcripts encoding mitochondrial proteins [106]. It regulates the expression of mitochondrial proteins functioning in key metabolic pathways in a posttranscriptional fashion. It promotes both mRNA stability and their translation [107]. In line with this, the *Drosophila* orthologue Clueless tethers ribosomes to the outer mitochondrial membrane via interaction with the TOM complex subunit Tom20 [108]. It remains to be determined what kind of G3BP1-containing granule hitchhikes on endolysosomes and whether mRNAs encoding mitochondrial proteins are coming along for the ride (Figure 2C).

## 5. Interplay between Cellular Signaling and Local Translation

While transcription of all nuclear-encoded mitochondrial proteins is regulated on a global scale in the cell body via the PGC-1α-NRF-1/2-TFAM pathway, the discovery of local translation may act as contributor to mitochondrial diversity within one cell. However, so far very little is known about the signaling pathways regulating mitochondrial biogenesis at the translational level in axons far away from the cell body. It is very likely that neurons have developed their own, neuron-specific mechanisms for regulating local mitochondrial biogenesis that are precisely adapted to their complex architecture and energetic demands.

Based on recent findings, it seems evident that protein translation preferentially occurs at contact sites between different organelles. Both early endosomes and endolysosomes have already been identified as translational platform for mitochondrial proteins in neurons [84,97]. Interestingly, both mTORC1 and AMPK, which play an important role in regulating mitochondrial biogenesis, are recruited to endolysosomes for activation [109,110,111,112]. One scenario that activates mTOR locally is axonal injury, which indeed leads to upregulation of local translation in an mTORC1-dependent manner, including the *mTOR* transcript itself [113]. *mTOR* mRNA has been shown to be transported into axons by the RBP nucleolin [113]. Given the observed effect of mTORC1 on mitochondrial protein translation [45], it is possible that local translation of mitochondrial transcripts is also upregulated upon axonal injury. This will support local ATP generation, which is critical to promote axon regeneration and neuronal survival [114].

Similarly, upon mitochondrial damage, stabilization of the PINK1 kinase on the outer mitochondrial membrane promotes translation of nuclear-encoded respiratory chain complex mRNAs, which are localized to the outer mitochondrial membrane in a PINK1/Tom20-dependent manner [115]. This leads to an increase in local mitochondrial biogenesis in the vicinity of damaged organelles, potentially representing an attempt to rescue the mitochondrial damage preceding activation of mitophagy by PINK1 activity. Given the fact that the *Pink1* mRNA is locally translated and can get activated in axons [85,116], this local feedback loop may also be operational in axons and represent a mechanism to homeostatically boost local mitochondrial biogenesis upon mitochondrial stress. Furthermore, other proteins have been reported to localize mRNAs to mitochondria. The outer mitochondrial membrane proteins A-kinase anchoring protein 1 (AKAP1) in humans and MDI in *Drosophila* recruit transcripts to the mitochondrial surface [117,118,119] and are involved in the selection mechanism that prevents transmission of deleterious mtDNA mutations in *Drosophila* oocytes [120]. Interestingly, this is again dependent on PINK1, which phosphorylates the translation stimulator La-related protein (Larp) to selectively inhibit local protein synthesis on defective mitochondria thereby limiting the replication of their mtDNA [120].

Interestingly, CLUH-dependent RNA granules have also been shown to function as signaling hubs, which control mTORC1 recruitment and activation as well as the function of other RBPs such as G3BP1 and 2 as mentioned above. Upon starvation, CLUH inhibits mTORC1 and stimulates mitochondrial turnover via mitophagy [105]. The function of CLUH in axons is not fully known yet. It is, however, very likely that CLUH plays a critical role due to its function in mitochondrial maintenance by controlling local translation and mitophagy.

Finally, axonal branching and growth as well as axonal regeneration are also associated with local translation of mitochondrial proteins. These processes require huge amounts of energy and are supported by local ATP production via axonal mitochondria [103,121,122,123,124]. A recent study showed that the nuclear-encoded mitochondrial translation initiation factor 3 (mtIF3) is locally translated in developing axons and promotes axonal translation of mitochondrial proteins thereby supporting axonal growth [125]. In addition to axonal growth, synaptic transmission is also an energy-consuming process. Interestingly, synaptic signaling stimulates AMPK activity due to decreased ATP levels [126,127]. Given the huge amounts of energy consumed by synaptic transmission [128], one may speculate that presynaptic AMPK activation can have two effects: Firstly, it may exert a global activation of the PGC-1α response leading to increased mitochondrial biogenesis on a global scale. In line with that, depolarization-induced AMPK activation results in increased PGC-1α, NRF-1 and TFAM levels as well as increased ATP production in primary rat visual cortical neurons [129]. However, it still remains to be determined if AMPK activity limited to individual synapses would create a signal that is able to propagate or be transported all the way to the cell soma to elicit a global transcriptional response. Secondly, local AMPK activation may restrict local synthesis of mitochondrial proteins via inhibition of mTORC1-dependent protein translation during these times of high ATP demand. After a period of acute synaptic activity, ATP levels can recover, resulting in decreased AMPK activity and potentially mTORC1 stimulation. Subsequent local translation of mitochondrial proteins may contribute to the repair and replenishment of mitochondria in axons leading to a complete restoration of local ATP levels. However, it has been shown that synaptic activity induces translation of mitochondrial proteins [130] but it remains to be determined if this increase occurs pre- or post-synaptically. Interestingly, chronically elevated intracellular Ca^2+^ levels, as may be the case during sustained synaptic signaling, have been shown to inhibit AMPK via activation of the protein phosphatase 2A in muscle cells [131,132], which would allow for mTORC1-dependent protein synthesis. Taken together we propose a model in which, during acute synaptic signaling, AMPK activation may reduce local mitochondrial protein synthesis to preserve energy. This would be in contrast to sustained activity that may activate axonal translation analogous to the post-synaptic compartment [133]. However, it remains to be determined if local AMPK activation upon acute synaptic activity indeed inhibits local mTORC1 signaling in areas with ATP shortage.

## 6. Importance of Mitochondrial Biogenesis in Neuronal Health

Mitochondrial maintenance including precise regulation of mitochondrial biogenesis plays a crucial role in neuronal health. This becomes apparent in the fact that the PGC-1α-NRF-1/2-TFAM pathway is downregulated in many neurodegenerative diseases including Parkinson’s disease (PD), Huntington’s disease (HD) and Alzheimer’s disease (AD) [11,134,135]. In PD, which is characterized by dopaminergic neurons loss in the substantia nigra, mtDNA levels [136] as well as the expression levels of PGC-1α and its target genes, such as NRF-1, are reduced [137,138]. The parkin interacting substrate (PARIS), whose levels are controlled by phosphorylation via PINK1 and ubiquitination via the E3 ubiquitin ligase parkin [139], has been shown to repress the transcription of PGC-1α by binding to its promoter [138]. Interestingly, conditional knockout of parkin or overexpression of PARIS results in the selective loss of dopaminergic neurons in the substantia nigra. Parkin or PGC-1α coexpression can prevent the PARIS-induced dopaminergic neuron loss [138]. In HD, a mutation in the huntingtin gene leads to aggregation of the mutant huntingtin protein and neurodegeneration in the striatum. Interestingly, mutant huntingtin has been shown to associate with the promoter of PGC-1α, thereby repressing its transcription and leading to reduced PGC-1α levels [140]. Accordingly, overexpression of PGC-1α in cultured striatal neurons and transgenic HD mice is neuroprotective [140]. In hippocampal tissue from AD patients and M17 cells overexpressing an AD-causing amyloid precursor protein (APP) mutant, the number of mitochondria is reduced [141,142]. Furthermore, the levels of PGC-1α, NRF-1/2 and TFAM are decreased, suggesting impaired mitochondrial biogenesis as cause for the reduced mitochondrial number [143]. Accordingly, overexpression of PGC-1α can reverse the mitochondrial biogenesis defect. Apart from PGC-1α, phospho-CREB levels are also reduced in the cells overexpressing APP. Interestingly, cAMP can rescue the expression of phospho-CREB and PGC-1α, while inhibition of PKA prevents this effect [143]. This indicates that the cAMP/PKA/CREB pathway plays an important role in controlling PGC-1α expression in this AD model. As mentioned above, insulin is also a critical regulator of mitochondrial biogenesis. Besides hampered mitochondrial biogenesis and function, AD has also been associated with impaired insulin signaling [144,145,146]. Interestingly, studies have shown that intranasal delivery of insulin results in improved cognitive performance in AD patients [147,148,149,150]. Given the supportive role mitochondria perform during synaptic plasticity [151], insulin-induced mitochondrial biogenesis may contribute to the positive effects of intranasal insulin in these patients.

Apart from defects in the global transcriptional regulation of mitochondrial biogenesis via the PGC-1α-NRF-1/2-TFAM pathway, it has been shown that mitochondrial health is also severely impaired when transport and local translation of nuclear-encoded mitochondrial transcripts are disturbed [152,153], with dire consequences for cellular and organismal health. Loss of local translation of mitochondrial transcripts, such as *laminb2* and *bclw*, as well as their RBP SFPQ, has been shown to result in axon degeneration [86,93,94], while dysregulation of *Cox4* mRNA transport into axons of the mouse forebrain leads to increased reactive oxygen species levels accompanied by an anxiety- and depression-like phenotype [154]. Furthermore, the mitophagy protein PINK1, whose transcript is hitchhiking on mitochondria along axons [85] and which modulates the synthesis of other mitochondrially associated transcripts [115], is mutated in familial forms of PD [155]. Axons of the predominantly affected cell type in PD, dopaminergic neurons, show a unique architecture. They belong to the most branched neurons with a tremendously complex arborization [156]. Accordingly, mRNA transport along axons and proper mitochondrial maintenance, including biogenesis and mitophagy, is of particular importance in dopaminergic neurons and one of the reasons why they may be more susceptible to PD.

Additionally, defects in transport and translation at mito-endolysosomal contact sites may impact the local availability of mitochondrial proteins. Loss of the FERRY complex, which tethers nuclear-encoded mitochondrial transcripts and ribosomes to early endosomes, has been shown to severely compromise brain development and function resulting in intellectual disability and epilepsy [97], which may be linked to defects in local translation of mitochondrial proteins. Other proteins involved in mRNA hitchhiking on endolysosomes are found mutated in neurodegenerative diseases, including Rab7 and ANXA11. Missense mutations in Rab7, a small GTPase involved in transport of late endosomes, cause Charcot-Marie-Tooth type 2B (CMT2B) [157]. As a consequence, impaired local translation of mitochondrial proteins on axonal endosomes has been observed in neurons carrying disease relevant mutations. This results in compromised mitochondrial function and axonal viability [84]. ANXA11, on the other hand, has been shown to be associated with amyotrophic lateral sclerosis (ALS) [158,159]. Mutations in ANXA11 disrupt the interaction between mRNPs and lysosomes, thereby impairing mRNA transport along axons in ALS [96]. Since RBPs play a critical role in proper axonal localization of mRNAs, it is not surprising that mutations or loss of RBPs have been linked to neurological diseases [160]. For instance, mutations in the TAR-DNA-binding-protein 43 (TDP-43) also result in familial forms of ALS as well as frontotemporal dementia [161]. ALS-associated mutations in TDP-43 have been demonstrated to directly impair axonal mRNA transport [162,163] due to the formation of RNP condensates, which affect mRNA localization and translation [164]. Furthermore, a recent study shows that pathological TDP-43 condensates primarily affect local synthesis of nuclear-encoded mitochondrial proteins in axons and synapses of motor neurons. This leads to impaired mitochondrial function and consequently neuromuscular junction degeneration [165]. These results suggest that the localization of mitochondrial transcripts as well as axonal translation are crucial determinants of neuronal health.

Taken together, mitochondrial biogenesis is impaired in several neurodegenerative diseases, both at the global level of PGC-1α and at the level of controlled transport and local translation of mitochondrial transcripts. Targeting mitochondrial biogenesis at both levels may result in promising therapeutic strategies. 

## 7. Conclusions

In several studies, it has become evident that transcriptional upregulation does not necessarily lead to increased protein levels [59,60,61] as these processes are regulated by different signaling pathways. This is also the case for mitochondrial biogenesis. While the cell-intrinsic molecules including the AMP/ATP ratio, the NAD^+^/NADH ratio and Ca^2+^ levels control transcription via AMPK and the PGC-1α-NRF1/2-TFAM pathway, the cell-extrinsic molecule insulin mainly controls translation of mitochondrial proteins via the AKT/mTORC1 pathway. Due to the separate regulation at the transcriptional and translational level, the cell can respond differently depending on the nature of the stimulus, thereby allowing for short- or long-term adaptation in the regulation of mitochondrial biogenesis. Cell-intrinsic activation of AMPK promotes transcription while potentially downregulating translation of mitochondrial proteins due to mTORC1 inhibition. Although very little is known about the regulation of mitochondrial biogenesis in neurons, this regulatory pathway may be of particular importance given the mitochondrial dysfunction observed in neurodegenerative diseases. Understanding the regulatory pathways controlling neuronal mitochondrial biogenesis at all levels and the critical contribution of axonal local translation at organellar contact sites will not only provide new insights into the complexity of mitochondrial maintenance in highly polarized cells such as neurons but may also result in new therapeutic targets for various neurodegenerative disorders.

## Figures and Tables

**Figure 1 biomolecules-12-01595-f001:**
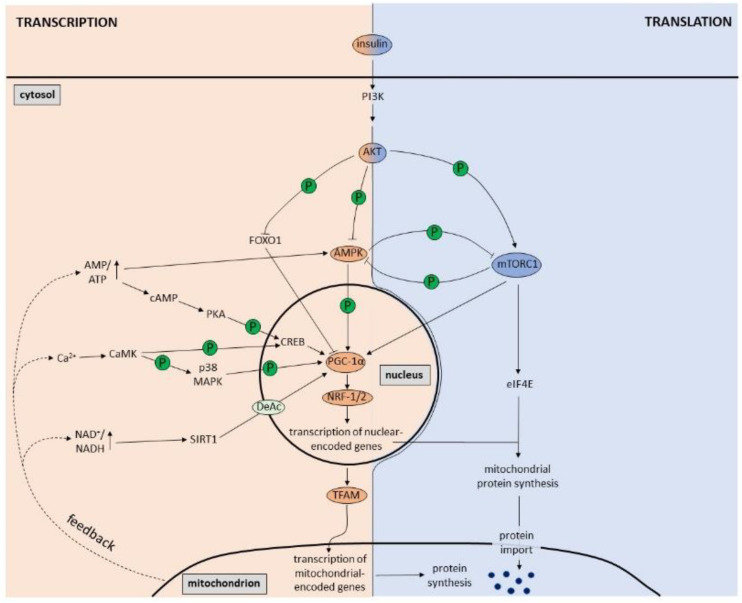
Regulation of mitochondrial biogenesis at the transcriptional and translational level. At the transcriptional level, the PGC-1α-NRF-1/2-TFAM pathway is the master regulator of mitochondrial biogenesis. Increased AMP/ATP ratio, NAD^+^/NADH ratio and Ca^2+^ levels, which are part of mitochondrial feedback mechanisms within the cell, result in activation of AMPK, PKA, SIRT1 and CaMK that in turn lead to PGC-1α stimulation. Once activated, PGC-1α promotes transcription of nuclear-encoded mitochondrial genes via NRF-1/2 and transcription of mitochondrial-encoded genes via expression of TFAM. At the translational level, the insulin-induced PI3K/AKT/mTORC1 pathway plays a major role in mitochondrial biogenesis. Via activation of eIF4E, mTORC1 increases the translation of nuclear-encoded mitochondrial proteins, which are imported into mitochondria. AKT, however, also has an effect on transcription via inhibition of FOXO1 and consequent activation of PGC-1α. Furthermore, AMPK is also a substrate of AKT, which is inhibited by AKT-induced phosphorylation. Finally, AMPK and mTORC1 can inhibit each other by direct phosphorylation.

**Figure 2 biomolecules-12-01595-f002:**
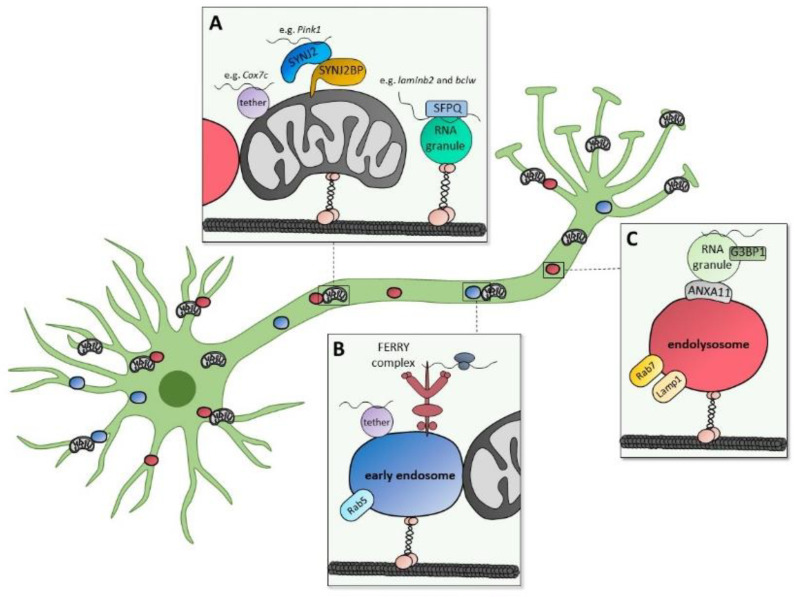
Transport of nuclear-encoded mitochondrial transcripts in neurons. Motor proteins actively transport RNA granules, mitochondria and Rab5-positive early endosomes as well as Rab7- and LAMP1-positive endolysosomes along axons. In this way, nuclear-encoded mitochondrial transcripts that are either part of RNA granules or tethered to the organelles reach distal parts of the neurons allowing for local translation. (**A**) The mitochondrial *laminb2* and *bclw* transcripts are transported in RNA granules via binding to their RBP SFPQ. *Cox7c* and *Pink1* mRNA are tethered to mitochondria. The tethering complex for *Pink1* mRNA is composed of SYNJ2BP and SYNJ2. (**B**) Several mRNAs are transported with early endosomes. The FERRY complex localizes to early endosomes and interacts both with the translation machinery and mRNAs that are enriched for nuclear-encoded mitochondrial genes. (**C**) Endolysosomes transport G3BP1-containing RNA granules using ANXA11 as a tether.

## Data Availability

Not applicable.

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
