# Peer review of "Metabolic Regulation of Mitochondrial Protein Biogenesis from a Neuronal Perspective"

_biomolecules, 2022, doi:10.3390/biom12111595_

Round 1

Reviewer 1 Report

In this review paper, Hees et al discuss the regulatory processes facilitating the biogenesis of mitochondrial proteins in neuronal cells, focusing on signalling pathways and local translation in axons. The review is well written and gives a thorough view of the topic with the addition of a few future perspectives. I will advise you to accept this manuscript for publication upon a few minor points:

1.     There needs to be a general distinction between metabolic pathways leading to the generation of new mitochondria (PGC1α related), and pathways related to rejuvenation and recycling (such as mitophagy processes). Although there is close interaction between the two, I would suggest including a short description of known signalling pathways in neurons related to the degradation of mitochondria. This is especially relevant as pink1, a key player in the second part of the review, is tightly connected to degradation rather than to the biogenesis of new mitochondria.

2.     Other than synaptic transmission, axon growth is also an energy-consuming process in neurons, which is related to mitochondrial activity. It is important to discuss how the local translation of nuclear-encoded mitochondrial proteins can also influence axon growth, as was recently demonstrated by Lee et al (2022) for mtIF3 protein. 

3.     Better phrasing should be applied in a few instances, such as lines 86 and 417

Author Response

We would like to thank the reviewer for the helpful suggestions. We have added a paragraph on mitophagy as a mitostatic mechanism and provide further reading suggestions on the metabolic regulation of this process. We also added axon growth and the local translation of mtIF3 to the description of local mitochondrial biogenesis, as well as corrected the phrasing in the lines mentioned.

Reviewer 2 Report

In the manuscript entitled “Metabolic regulation of mitochondrial protein biogenesis in neurons” authors review the knowledge mainly on regulation of expression of nuclear genes encoded in nucleus. Expression of these genes is regulated on the level of transcription, transport to the appropriate neuron subcompartment (e.g. synapse), translation and transport to the organelle. The review is well written and I do not suggest any corrections or improvement. As far as I can judge, it is also written in good english. 

Author Response

We thank this reviewer for this kind assessment.

Reviewer 3 Report

Neurons have unique cell morphology along with high demand of energy. To fit the demand, different strategies of regulating mitochondrial activity are utilized in the neurons. The manuscript aimed to review the regulation of mitochondrial biogenesis in neuron cells. The basics of mitochondria metabolism and protein production were described. The possible mechanisms of mitochondria mRNA and assembly in axon was discussed. In the end, the neurological diseases related to mitochondria were reviewed. The figures in the manuscript are well-prepared with high quality. Readers will benefit from the information collected.

Comments:

The title of the manuscript is " Metabolic regulation of mitochondrial protein biogenesis in neurons.". However, most of manuscript is based on mitochondria. It is the reviewer’s suggestion that the author could provide more neuron-specific mitochondrial features to emphasize topics in the title.

Author Response

We thank this reviewer for this assessment. We would have loved to include more “neuron-specific” details, yet as we already mention in the text, most research on the mechanisms regulating mitochondrial biogenesis have not been studied in neurons directly. To make this clearer to future readers, we therefore altered the title to “Metabolic regulation of mitochondrial protein biogenesis from a neuronal perspective” to reflect that not only neuronal studies will be reviewed.